# STAT3 Mediated miR-30a-5p Inhibition Enhances Proliferation and Inhibits Apoptosis in Colorectal Cancer Cells

**DOI:** 10.3390/ijms21197315

**Published:** 2020-10-03

**Authors:** Chun-Chia Cheng, Bi-Ling Yang, Wen-Chao Chen, Ai-Sheng Ho, Zong-Lin Sie, Hsin-Chi Lin, Chun-Chao Chang

**Affiliations:** 1Radiation Biology Research Center, Institute for Radiological Research, Chang Gung University/Chang Gung Memorial Hospital, Linkou 333, Taiwan; cccheng.biocompare@gmail.com (C.-C.C.); zonlins@gmail.com (Z.-L.S.); 2Division of Gastroenterology, Cheng Hsin General Hospital, Taipei 112, Taiwan; linfrank24@gmail.com (B.-L.Y.); aisheng49@gmail.com (A.-S.H.); ogi861047@yahoo.com.tw (H.-C.L.); 3Division of Gastroenterology and Hepatology, Department of Internal Medicine, Taipei Medical University Hospital, Taipei 110, Taiwan; garychen0716@yahoo.com.tw; 4Division of Gastroenterology and Hepatology, Department of Internal Medicine, School of Medicine, College of Medicine, Taipei Medical University, Taipei 110, Taiwan

**Keywords:** colorectal cancer, STAT3, miR-30a-5p, regorafenib, HSPA5

## Abstract

Signal transducer and activator of transcription 3 (STAT3), a transcriptional factor involved in tumorigenesis and cancer stemness formation, contributes to drug resistance in cancer therapies. STAT3 not only mediates gene transcription but also participates in microRNA suppression. This study identified a STAT3-downstream micro RNA (miRNA) involved in drug resistance against regorafenib in colorectal cancer stem-like tumorspheres. Small RNAseq was used to investigate differential microRNAs in colorectal cancer cell-derived tumorspheres and in a STAT3-knockdown strain. The miRNA-mediated genes were identified by comparing RNAseq data with gene targets predicted using TargetScan. Assays for detecting cell viability and apoptosis were used to validate findings. The formation of colorectal cancer stem-like tumorspheres was inhibited by BBI608, a STAT3 inhibitor, but not by regorafenib. Additional investigations for microRNA expression demonstrated an increase in 10 miRNAs and a decrease in 13 miRNAs in HT29-derived tumorspheres. A comparison of small RNAseq results between tumorspheres and HT29shSTAT3 cells revealed the presence of four STAT3-mediated miRNAs in HT29-derived tumorspheres: hsa-miR-215-5p, hsa-miR-4521, and hsa-miR-215-3p were upregulated, whereas miR-30a-5p was downregulated. Furthermore, hsa-miR-4521 was associated with poor overall survival probability, and miR-30a-5p was associated with better overall survival probability in patients with rectum cancer. Comparisons of RNAseq findings between HCT116- and HT29-derived tumorspheres revealed that HSPA5 were mediated by the STAT3-miR-30a-5p axis, which is overexpressed in colorectal tumorspheres associating to anti-apoptosis. In addition, the transfection of miR-30a-5p and inhibition of HSPA5 by HA15 significantly reduced cell viability and increased apoptosis in HT29 cells. In conclusion, a STAT3-miR-30a-5p-HSPA5 axis was observed against regorafenib-mediated apoptosis in colorectal cancer tumorspheres. The expression of miR-30a-5p was repressed by STAT3; in addition, HSPA5 was identified as the target gene of miR-30a-5p and contributed to both tumorsphere formation and anti-apoptosis.

## 1. Introduction

Colorectal carcinoma (CRC) is a lethal disease with a high prevalence worldwide; it is reported as the third most common cancer globally with a death rate of approximately 10% [1]. Common CRC treatments include surgical resection, radiotherapy, chemotherapy, targeted therapy, and immunotherapy. In addition to antibodies against angiogenesis and the epidermal growth factor (EGF) receptor (EGFR), only one targeted therapy against CRC, namely regorafenib, has been approved for use in clinical practice. Regorafenib has been approved for use in patients with metastatic colorectal cancer, resulting in a 26% disease control rate (complete response, partial response, or stable disease lasting at least 6 weeks) [2], but the antitumor activity of regorafenib is restricted to intrinsic and acquired resistance in CRC.

Regorafenib could increase signal transducer and activator of transcription 3 (STAT3) phosphorylation in colorectal cancer cells, HCT116 and HT29 [3]. Alternatively, another study reports that regorafenib significantly inhibited intrinsic STAT3 phosphorylation in hepatocellular carcinoma cell lines due to increased SHP-1 phosphatase activity, resulting in cell apoptosis [4]. HCT116 and HT29 cells demonstrate high cancer stemness due to the overexpression of CD133 and CD44 [5]; by contrast, CD133 is absent in hepatocellular carcinoma cells [6]. Furthermore, regorafenib inhibits the activation of receptor tyrosine kinases (RTKs), including FGFR, PDGFR, VEGFR, TIE-2, RET, and KIT, resulting in a reduction in downstream signaling [7]. Based on these findings, we assumed that STAT3, induced by regorafenib under stressful conditions in CD133-positive HCT116 and HT29 cells, plays a vital role in maintaining both tumor survival and cancer stemness. To the best of our knowledge, colorectal cancer cells demonstrate high stemness and express either CD133 or CD44 [8]. In addition, constitutive STAT3 activation was reported to be involved in the formation of CRC-derived cancer stem-like tumorspheres [9]. Therefore, inhibition of STAT3 significantly abolishes the formation of cancer stem-like tumorspheres. Because STAT3 phosphorylation in CRC cell lines with strong cancer stemness characteristics may respond to regorafenib for tumor survival. In the present study, we investigated the role of STAT3 in drug resistance against regorafenib and determined the mechanism involved.

STAT3 is a transcriptional factor that responds to gene expression. However, accumulated evidence indicates that STAT3 can repress the expression of microRNAs (miRNAs), short noncoding RNAs that repress post-transcriptional gene expression by binding to complementary sequences on the untranslated regions of targeted mRNAs, thus alternatively regulating gene expression. STAT3 can not only increase the expression levels of miRNAs in tumor cells [10,11] but also repress their levels (e.g., miR-200c and miR-145-5p) during cancer stem cell (CSC) growth [12,13]. Epigenetic regulation is considered as a strategy to rapidly respond to microenvironment stresses in tumors. The possible mechanism to repress miRNA expression by STAT3 is by its interaction with and induces gene silencer G9a to methylate on H3K9. Therefore, this study focused on the identification of STAT3-mediated miRNAs and their role in drug resistance against regorafenib. The miRNA expression profiles between HT29-derived stem-like tumorspheres and HT29shSTAT3 cells were compared.

## 2. Results

### 2.1. Colorectal HCT116 and HT29 Tumor Cells Highly Expressed CD133

HCC116-and HT29-derived tumorspheres were cultured in serum-free DMEM medium containing individual 20 ng/mL of epidermal growth factor (EGF), basic fibroblast growth factor (bFGF), hepatocyte growth factor (HGF), and interleukin (IL)-6, which are sensitive to STAT3 inhibitors [9]. Stemness markers, including CD133 and LGR5, were detected at first; it indicated that HCT116 was a CD133-positive cell line (97.5% CD133 expression); however, only 0.93% of LGR5 expression was observed in these cells (Appendix A). The corresponding values in HT29 cells were as follows: 74.8% CD133 and 1.8% LGR5 expression (Appendix A).

### 2.2. HCT116- and HT29-Derived Tumorspheres Were More Sensitized to BBI608 Than Regorafenib

The IC50 value of regorafenib was 10.6 μM and 19.1 μM in HCT116 and HT29 cells, respectively (Figure 1A). Moreover, BBI608, a STAT3 inhibitor, was used as a positive control to inhibit the formation of HCT116- and HT29-derived tumorspheres since STAT3 contributes to CD133 expression in tumors [14,15]. But, there was no significant change in BBI608-treated T29 cells (data not shown). Despite that, the IC50 value of BBI608 was 2.6 μM and 26.6 μM in HCT116 and HT29 cells, respectively (Figure 1A). We found that only BBI608 inhibited intrinsic STAT3 phosphorylation in HT29 cells, but not for regorafenib (Figure 1B). Meanwhile, regorafenib increased STAT3 phosphorylation in 24 h treatment (Figure 1B). To investigate the anti-tumor effect for the two selected inhibitors, apoptosis was detected after 24 h treatment. BBI608 induced 8.64% and 16.4% apoptosis in HCT116 and HT29 cells, respectively whereas the corresponding proportions observed with regorafenib were 2.61% and 1.67%, respectively (Figure 2A). Quantification of apoptosis is shown in Figure 2B, it revealed that BBI608 induced remarkable apoptosis in colorectal cancer cells. Interestingly, the IC50 dosages of BBI608 significantly inhibited the formation of HCT116- and HT29-derived tumorspheres when compared with regorafenib (Figure 2C). These results revealed that HCT116- and HT29-derived tumorspheres were more sensitized to BBI608 than regorafenib.

### 2.3. Identification of STAT3-Mediated miRNAs in HT29-Derived Tumorspheres

miRNAs can regulate gene expression and contribute to tumor survival and drug resistance. In the present study, 10 upregulated and 13 downregulated miRNAs were detected in HT29-derived tumorspheres compared with HT29 cells (Figure 3A). Moreover, 21 miRNAs were upregulated and 48 were downregulated in HT29shSTAT3 cells compared with HT29shLuc cells (Figure 3B). Thus, three miRNAs regulated by STAT3, miR215-5p, miR4521, and miR215-3p, were overexpressed in HT29-derived tumorspheres (Figure 3C); alternatively, the expression level of miR30a-5p, which was also regulated by STAT3, was decreased in HT29-derived tumorspheres (Figure 3D). miRNAs detected in this study are listed in Figure 3E. The Kaplan–Meier plotter was used to identify miRNAs associated with survival probability in patients with colorectal cancer based on the database of findings of patients with rectum adenocarcinoma (n = 160, https://kmplot.com/analysis/index.php?p=service&cancer=pancancer_mirna). Results revealed that high levels of miR4521 and miR30a-5p expression were associated with poor and better probabilities, respectively (Appendix A).

### 2.4. STAT3 Suppressed miR-30a-5p to Inhibit Apoptosis in Colorectal HT29 Cancer Cells

miR-30a-5p expression levels were downregulated in HT29-derived tumorspheres when compared with parental HT29 cells (Figure 4A). Furthermore, the expression level of miR-30a-5p increased in HT29shSTAT3 cells compared with HT29shLuc cells (Figure 4B), indicating that STAT3 could reduce miR-30a-5p expression levels. Growth factors (EGF, bFGF, IL6, and HGF combined) added during the formation of tumorspheres inhibited miR-30a-5p expression in HCT116 and HT29 cells (Figure 4C). Moreover, STAT3 could bind to the promotor of miR-30a-5p in HT29 cells detected using ChIP technique (Figure 4D). To investigate the function of miR-30a-5p in colorectal cancer, miRNA was transfected into HT29 cells, and both cell viability and apoptosis were measured. The initial use of miRNA-FAM (stable negative control conjugated with FAM) revealed a transient miRNA transfection efficiency of 21.7% in HT29 cells (Figure 4E). Subsequently, the transfection of a miR-30a-5p mimic significantly reduced cell viability (Figure 4F) and increased apoptosis in HT29 cells (Figure 4G).

### 2.5. HSPA5 was the Downstream Target of miR-30a-5p and Was Involved in Anti-Apoptosis

RNAseq was used to identify differentially expressed genes in HCT116- and HT29-derived tumorspheres, and the differential genes were compared with miR-30a-5p targets predicted by TargetScan (http://www.targetscan.org/vert_72/). A total of 654 and 101 genes were upregulated and 840 and 90 genes were downregulated in HCT116-and HT29-derived tumorspheres, respectively, when compared with parental cells (Figure 5A). The overlapping of 32 upregulated genes was observed between HCT116 and HT29 CSCs (Figure 5B). HSPA5 was found to be associated with anti-apoptosis based on the PANTHER BP (Figure 5C). Consequently, the selected overexpressed driver genes, namely HSPA5, SFN, MAFF, ATF3, PPARD, MSMO1, HMGCS1, NDRG1, MVD, and HERPUD1, were validated. Furthermore, HSPA5, SFN, MSMO1, HMGCS1, NDRG1, and MVD were significantly increased in HT29-derived tumorspheres in the qPCR analysis (Figure 5D). By contrast, comparisons between 32 upregulated genes and the target genes of miR-30a-5p revealed three genes, TSC22D2, CALB2, and HSPA5 (Figure 5E), that were increased in HT29-derived tumorspheres (Figure 5F) and growth factor-treated cells (Figure 5G). The levels of TSC22D2, CALB2, and HSPA5 decreased in HT29shSTAT3 cells compared with HT29shLuc cells (Figure 5H). Since HSPA5 was associated with anti-apoptosis, as predicted by NetworkAnalyst. Furthermore, the protein levels of HSPA5 in HT29shSTAT3 cells were lower than those in HT29shLuc (Figure 5I) and BBI608 reduced HSPA5 expression (Figure 5I). HSPA5 increased in HT29-derived tumorspheres (Figure 5J) but decreased in miR-30a-5p mimic-transfected HT29 cells (Figure 5K).

### 2.6. Inhibition of HSPA5 Reduced Cell Viability and Increased Apoptosis in Colorectal Cancer-Derived Tumorspheres

According to the Kaplan-Meier plot in rectum adenocarcinoma (https://kmplot.com/analysis/index.php?p=service&cancer=pancancer_rnaseq), the overexpression of HSPA5 was associated with poor progression-free survival (PFS) probability (Appendix A). The mRNA expression data for STAT3 and HSPA5 by RNAseq in CRC patinets (n = 244) was obtained and investigated from cBioPortal for Cancer Genomics website. The result indicated that STAT3 was positively associated with HSPA5 in patients with colorectal cancer (r^2^ = 0.12, *p* < 0.0001, Figure 6A). We found that HA15, an HSPA5 inhibitor, significantly inhibited cell viability in HCT116 and HT29 cells (16.6 μM and 31 μM IC50, respectively; Figure 6B) and increased apoptosis in HCT116 and HT29 cells with IC50 concentrations (Figure 6C). In addition, HA15 significantly diminished the formation of HCT116- and HT29-derived tumorspheres (Figure 6D).

## 3. Discussion

This study demonstrated that the downregulation of miR-30a-5p in colorectal cancer-derived tumorspheres is regulated by STAT3. The targeting of STAT3 by BBI608 significantly diminished tumorsphere formation and induced apoptosis in HCT116 and HT29 cells. The transfection of a miR-30a-5p mimic reduced cell viability and increased apoptosis. Furthermore, HSPA5 was identified as the target gene of miR-30a-5p and contributed to both tumorsphere formation and anti-apoptosis.

STAT3 mediates tumor proliferation, invasion, anti-apoptosis, and stemness and acts as a therapeutic target. Multiple signals regulate STAT3 phosphorylation, including receptors, such as EGFR, gp130, vascular endothelial growth factor receptor (VEGFR), platelet-derived growth factor receptor (PDGFR), and insulin-like growth factor receptor (IGFR), and nonreceptors, such as src, Bcr-Abl, and BMX on Tyr705 [16]. Other stimulations that mediate the phosphorylation of STAT3 on Ser727 include p38MAPK, ERK, JNK, PKC, and mTOR [16]. Therefore, STAT3-upstream signals may be activated under regorafenib-induced stressful conditions [17,18], leading to STAT3 activation. Moreover, additive growth factors for tumorsphere formation, such as EGF and IL6, can phosphorylate STAT3 in cancer cells [9,13,19]. Thus, tumorspheres with constitutive STAT3 activation are expected to be resistant to regorafenib-inhibited cell growth. The targeting of STAT3 by BBI608 significantly induced apoptosis; the higher cytotoxicity of BBI608 than that of regorafenib mainly inhibited cell proliferation (Figure 2A,B).

A recent study showed that STAT3 mediates Bcl-2 expression and exerts anti-apoptotic effects on tumors [20]. However, we did not find any increase in Bcl-2 levels in HCT116- and HT29-derived tumorspheres in the current study (data not shown). By contrast, miR-30a-5p was found to be negatively regulated by STAT3 in this study. Moreover, miR-30a-5p was considered as a tumor suppressor in that higher miR-30a-5p resulted in better survival probability in patients with rectum adenocarcinoma (*p* = 0.059 but showing the trend). Details regarding the mechanism through which STAT3 reduced miR-30a-5p expression are warranted; nonetheless, this study showed that the downregulation of miR-30a-5p played a role in anti-apoptosis. STAT3 has been reported to both induce and repress microRNA expression [21,22,23,24]. One of the possible mechanisms might be the interaction of STAT3 with G9a and the epigenetic suppression of gene expression through methylation of H3K9, which thereby downregulate the expression levels of miR-200c and miR-145 [12,13]. Another methyltransferase (EZH2) is regulated by STAT3 and participates in anti-apoptosis in gastric cancer cells [25]. EZH2 contributes to H3K27 methylation and represses the expression levels of miR-200 [26], miR-30d [27], and miR-30a [28].

*miR-30a-5p* is considered a tumor suppressor gene, a novel biomarker, and a therapeutic target [29]. In addition, it has been shown to inhibit cell proliferation [29,30] and increase apoptosis by repressing the nuclear transport receptor KPNB1 [27,28,29]. However, KPNB1 is not included in the TargetScan database. Moreover, no significant changes in KPNB1 levels were observed in HCT116- and HT29-derived tumorspheres (data not shown). Alternatively, three miR-30a-5p-mediated target genes, TSC22D2, CALB2, and HSPA5, were upregulated in tumorspheres; HSPA5 was associated with anti-apoptosis, as predicted by NetworkAnalyst. In addition, higher HSPA5 levels resulted in worsen survival probability in patients with rectum adenocarcinoma (*p* = 0.12 but showing the trend). We further demonstrated that the targeting of HSPA5 by HA15 significantly inhibited cell viability and increased apoptosis and prevented the formation of colorectal cancer HT29-derived tumorspheres, thus indicating that the STAT3-miR-30a-HSPA5 axis contributed to anti-apoptosis in CRC.

The expression level of miR-4521 was increased in HT29-derived tumorspheres regulated by STAT3. According to the Kaplan–Meier plot, a high miR-4521 level was associated with poor survival probability, although the *p* value is 0.066, thus indicating its potential as an oncogene. However, a previous study indicated that miR-4521 actually decreased in tumors such as clear cell renal cell carcinoma as a tumor suppressor for targeting oncogene FAM129A [31]. Additionally, a decrease in the expression level of miR-4521 was reported in renal cell carcinoma cells [32]. Therefore, this miRNA was not further investigated in the current study.

*HSPA5*, also called Bip or GRP78, is a master regulator of unfolded protein response (UPR). The inhibition of this gene increases UPR, resulting in cell death due to the concomitant induction of autophagy and apoptosis. Under normal conditions, HSPA5 binds to and inactivates the ER stress sensors ATF-6, PERK, and IRE1. HSPA5 is highly expressed on the surfaces of various cancer cells. In addition to HSPA5, the levels of SFN, MSMO1, HMGCS1, NDRG1, and MVD were increased in HCT116- and HT29-derived tumorspheres. SFN (stratifin) participates in the development and progression of lung cancer [33] and stabilizes both EGFR and MET by interacting with ubiquitin-specific protease 8 (USP8) [34]. Extracellular acidic pH due to nutrient starvation activates SREBP2 to induce the expression of MSMO1, which contributes to tumor growth and malignancy [35]. HMGCS1 is upregulated in prostate cancer, and the knockdown of this gene inhibits cell viability in 22Rv1 cells [36]. NDRG1 has been reported as a tumor suppressor decreasing in colorectal cancer inhibits tumor proliferation through increasing p21 stability [37]. MVD is involved in mevalonate pathway for cholesterol biosynthesis but the role of this gene is not clear in tumors.

In this study, HCT116- and HT29-derived tumorspheres were found to be sensitive to BBI608 compared with regorafenib. Furthermore, the STAT3-miR-30a-HSPA5 axis contributed to anti-apoptosis and formation of tumorspheres. The targeting of STAT3 and HSPA5 with BBI608 and HA15, respectively, significantly increased apoptosis and diminished tumorsphere formation in colorectal HCT116 and HT29 tumor cells.

## 4. Materials and Methods

### 4.1. Cell culture and Tumorsphere Formation

HCT116 and HT29 cells gifted by the Institute of Nuclear Energy Research, Taiwan were free of mycoplasma, that was originally purchased from ATCC. The cells were cultured in Dulbecco’s modified Eagle’s medium (DMEM) with 10% fetal bovine serum and 1% penicillin–streptomycin. Tumorsphere formation and measurements were performed as described previously [38,39]. In this study, 20 ng/mL of combined EGF, bFGF, IL6, and HGF (GeneScript, Piscataway, New Jersey, USA) were used to culture tumorspheres for 7-days in a low-attachment plate without fetal bovine serum. The diameter of the tumorsphere was recorded through an inverted microscope.

### 4.2. Small RNAseq Profiling and Bioinformatics Analysis

Differentially expressed miRNAs were investigated using the small RNA digitalization analysis by sequencing and synthesis (Illumina, San Diego, California, USA). The expression levels of known and unique miRNAs in each sample were statistically analyzed and normalized using transcripts per million clean tags (TPMs). Common differential miRNAs between HT29-derived tumorspheres and parental HT29 cells, and HT29shSTAT3 and HT29shLuc were compared.

### 4.3. RNAseq Profiling and Bioinformatics Analysis

RNAseq analysis was performed to investigate differential genes in HCT116- (Appendix A) and HT29-derived tumorspheres (Appendix A) when compared with parental cell lines by using HiSeq 4000 with paired-end 150 bp sequencing. Genes with >1-fold change (log2) in expression levels in HCT116- and HT29-derived tumorspheres were consequently analyzed using NetworkAnalyst (http://www.networkanalyst.ca/). Pathway activations were selected and matched according to the PANTHER database.

### 4.4. Quantitative PCR

Procedures used for mRNA extraction and complementary DNA preparation have been described previously [38]. Cells were cultured in a 6-cm dish for mRNA extraction and harvested using 1 mL of TRIzol (Thermo Fisher Scientific, Massachusetts, USA). The solution was mixed with 200 μL of 1-bromo-3-chloropropane (Sigma-Aldrich, St. Louis, Missouri, USA), vortexed, and incubated for 5 min at room temperature. The supernatant was collected after 13,000-rpm centrifugation for 15 min at 4 °C. Isopropanol (500 μL) was added and incubated for 5 min at room temperature. The pellet was collected after 13,000-rpm centrifugation for 10 min at 4 °C. Subsequently, the pellet was incubated with 1 mL of 70% ethanol and centrifuged at 7500 rpm for 10 min at 4 °C. Furthermore, the mRNA pellet was dissolved in double-distilled water after air drying. To obtain cDNA, 1 μg of mRNA, 2 μL of random hexamers, and 10 μL of double-distilled water were mixed in a polymerase chain reaction (PCR) tube and incubated at 65 °C for 10 min, followed by cooling at 4 °C. The solution was mixed with 4 μL of buffer (5 ×), 0.5 μL of RNase, 2 μL of dNTP (2.5 mM), and 0.5 μL of reverse transcriptase, and it was consequently treated at 25 °C for 10 min, 50 °C for 1 h, and 85 °C for 5 min, followed by cooling at 4 °C. Quantitative polymerase chain reaction (qPCR) was performed using an SYBR Green system (Applied Biosystems, Foster City, CA, USA) according to manufacturer’s instruction. The primers are shown in Table 1.

### 4.5. Gene Knockdown

Gene knockdown was performed using a short-hairpin RNA (shRNA)-expression lentivirus system that contained the specific shRNA (target sequence of STAT3: GCACAATCTACGAAGAATCAA.) in the vector pLKO.1-puro generated in 293T cells as described in our previous study [38]. The plasmid and negative control (shLuciferase, sLuc) were purchased from National RNAi Core Facility of Academia Sinica, Taipei, Taiwan. For producing lentivirus, 293T cells (70% confluence) cultured in DMEM containing 10% FBS and 0.1% penicillin–streptomycin (6-cm dish) were transfected with 4 μg of shSTAT3 pLKO.1 vectors, 1 μg of the envelope plasmid pVSV-G, and 3.6 μg of the packaging plasmid pCMVΔR8.91. The plasmids were pre-incubated with 6 μL of JetPRIME for 20 min at room temperature and consequently added to 293T cells. The cultured medium was substituted with fresh DMEM containing 10% FBS and 1% of penicillin–streptomycin and incubated for 48 h. The virus solution was collected and stored at −80 °C. HT29 cells cultured in 80% confluence were infected with the prepared lentivirus (preincubated with 8 µg/mL of polybrene) for 24 h. The cells were then changed with DMEM medium containing 10% FBS, 1% penicillin–streptomycin, and 4 µg/mL of puromycin, which were harvested after obtaining stable cells.

### 4.6. Western Blotting

Western blotting was performed as described previously [38]. Specific antibodies against STAT3, pSTAT3, HSPA5, and GAPDH were purchased from Cell Signaling (Danvers, Massachusetts, USA). The cells were lysed in RIPA buffer containing 50 mM Tris-HCl (pH 7.4), 1% NP-40, 0.5% Na-deoxycholate, 0.1% sodium dodecyl sulfate (SDS), 2 mM ethylenediaminetetraacetic acid, 50 mM NaF, and 150 mM NaCl. The lysed proteins were mixed with 5× sample buffer [75 mM Tris-HCl, pH 6.8, 10% (v/v) glycerol, 2% SDS (w/v), 0.002% (w/v) bromophenol blue]. In total, 20 μg of each sample was analyzed through 10% SDS-polyacrylamide gel electrophoresis and then transferred onto Immobilon-P polyvinylidene fluoride (PVDF) membranes (Merck Millipore, Massachusetts, USA). These membranes were blocked with 5% skim milk for 1 h at room temperature, incubated with primary antibodies (1 μg/mL) overnight at 4 °C, and washed using Tris-buffered saline with 0.5% Tween-20. After washing, the PVDF membranes were incubated with horseradish peroxidase-conjugated secondary antibody (1 μg/mL) for 2 h at room temperature. The immunoreactive proteins were detected through an enhanced chemiluminescence kit (Bio-Rad, California, USA) coupled with an LAS-4000 mini device (Fujifilm, Tokyo, Japan). The results were quantified using ImageJ software.

### 4.7. Cell Viability

The WST-1 assay (Sigma-Aldrich, St. Louis, MO, USA) was used according to the manufacturer’s protocol to determine cell viability. The reagents, including BBI608, regorafenib, and HA15 (MedChemExpress, Monmouth Junction, NJ, USA) were used and added in HCT116 and HT29 cells for 48 h for cell viability detection.

### 4.8. miR-30a-5p Detection

A TaqMan advanced miRNA assay (Applied Biosystems, Waltham, MA, USA) was used to detect the expression of miR-30a-5p according to the manufacturer’s instructions.

### 4.9. Transfection of miR-30a-5p Mimic

JetPRIME was used to transfect miR-30a-5p mimic (UGUAAACAUCCUCGACUGGAAG) and negative control (NC, UUCUCCGAACGUGUCACGUTT) into HT29 cells. Here, NC conjugated with FAM was used for measuring the transfection efficiency. In brief, 5 μL of miRNAs (20 μM) was mixed with 200 μL of JetPRIME buffer, that was consequently added with 6 μl of JetPRIME solution. After incubation for 15 min at room temperature, the HT29 cells were added with the mixture for 24 h. The cells were collected after another 24 h.

### 4.10. Flow Cytometry

The selected cell lines HCT116 and HT29 in 100 μL of DMEM medium were incubated with anti-CD133-PE and anti-LGR5-PE, respectively (Biolegend, San Diego, CA, USA) for 0.5 h. Subsequently, the cells were added with 900 μL of phosphate-buffered saline (PBS). For apoptosis detection, a commercial kit containing Annexin V-FITC and Propidium Iodide was used (Strong Biotech Corporation, Taiwan). The tumor cells were treated with IC50 of inhibitor for 24 h, including BBI608, regorafenib, and HA15. HT29 cells transfected with miR-30a-5p mimic and negative control sequence were collected after 24 h. The cells were analyzed using a FACSCalibur Flow Cytometer (BD Bioscience, San Jose, CA, USA) after Annexin V-FITC and Propidium Iodide staining.

### 4.11. CIP

In brief, 1 × 10^7^ HT29 cells were treated with 20 ng/mL of EGF, FGF, HGF, and IL6 for 2 h. The HT29 cells were added and fixed by 1% formaldehyde for 5 min and then neutralized with 1 M of glycine. The fixed HT29 cells were suspended in 500 μL SDS lysis buffer (50 mM Tris-HCl, 10mM EDTA, 1% SDS, and adequate protease inhibitor cocktail (Sigma-Aldrich, St. Louis, MO, USA)) and sonicated for 24 cycles of 30-s pulse followed by 30-s rest on ice. To remove cell debris, the cells were centrifuged with 12,000× *g* for 10 min at 4 °C and the supernatant was collected. For immunoprecipitation, 500 μL supernatant was diluted with 1 mL IP dilution buffer (0.01% SDS, 1.1% Triton X-100, 16.7 mM Tris-HCl, 1.2 mM EDTA, 167 mM NaCl, and adequate protease inhibitor (Sigma-Aldrich, St. Louis, MO, USA)) and incubated with control IgG or anti-STAT3 antibody at 4 °C overnight. The antibodies were captured by 250 μg Dynabead-Protein A (Life technologies, Waltham, MA, USA) at room temperature for 30 min. To remove uncaptured protein and DNA fragments, the Dynabeads were washed sequentially using 1 mL of pre-chilled low salt wash buffer (0.1% SDS, 0.1% Triton X-100, 150 mM NaCl, 2 mM EDTA, and 20 mM Tris-HCl), 1 mL of high salt wash buffer (0.1 % SDS, 0.1% Triton X-100, 150 mM NaCl, 2 mM EDTA, and 20 mM Tris-HCl), 1 mL of IP wash buffer (0.5 M LiCl, 1% NP-40, 1% deoxycholic acid, and 100 mM Tris-HCl), and 1 mL of fresh prepared TE wash buffer (10 mM Tris-HCl and 1 mM EDTA). The Dynabeads were pulled down by a Sample Magnetic Rack to remove supernatant and pipetted up and down between each wash. To eluted DNA fragments, 200 μl Elution buffer was directly added to the Dynabeads and pipetted up and down for about 10 s, the samples were boiled for 10 min using, and supernatants were collected. For QPCR analysis, the DNA preparation was concentrated by a FavorPrep GEL/PCR Purification Mini Kit (Favorgen, Wembley, WA, Australia) and suspended with 50 μL distilled water. Then, 2.5 μL of each DNA sample was analyzed by real-time PCR using Fast SYBR Green Master Mix (Applied Biosystem, CA, USA) with primer pairs shown on Table 1. All reactions were run in triplicate. The primer sequences amplify the miR-30a-5p promoter predicted from PROMO (http://alggen.lsi.upc.es/cgi-bin/promo_v3/promo/promoinit.cgi?dirDB=TF_8.3).

### 4.12. Statistical Analysis

Statistical analyses were performed using GraphPad Prism V5.01 (GraphPad Software, Inc., California, USA). All analytical data with more than two groups were evaluated using analysis of variance followed by posthoc analysis with Bonferroni’s test. Student’s *t*-test was used to compare two groups. Moreover, *p* < 0.05 was used to indicate a statistically significant difference.

## Figures and Tables

**Figure 1 ijms-21-07315-f001:**
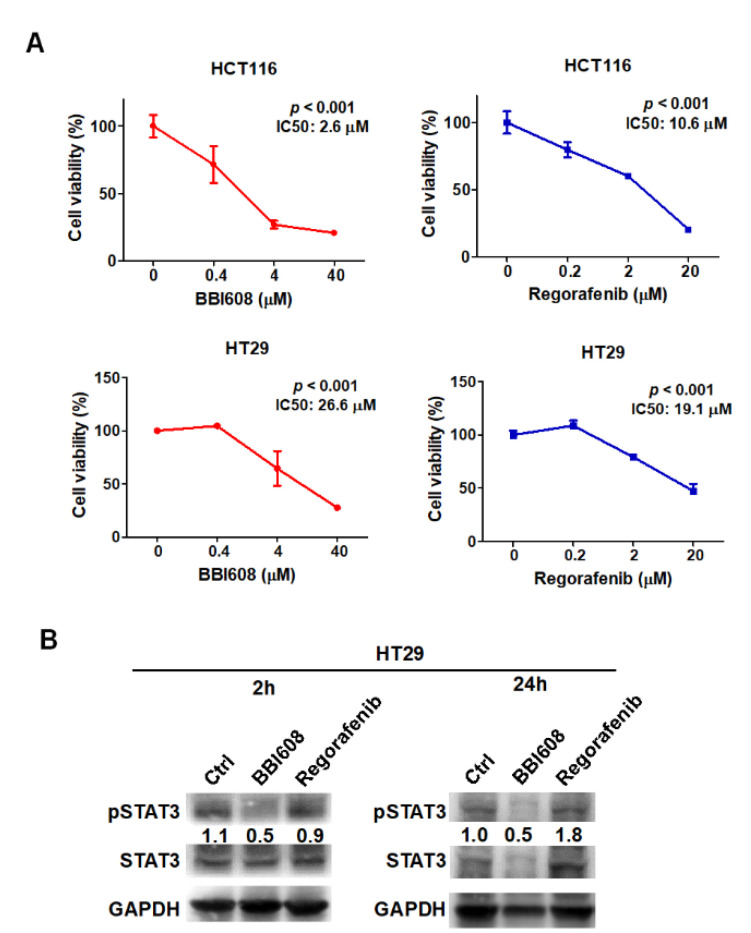
BBI608 inhibited phosphorylation of STAT3 in colorectal tumor cells. (**A**) BBI608 and regorafenib significantly reduced cell viability in HCT116 and HT29 cells. (**B**) BBI608 inhibited STAT3 phosphorylation but regorafenib stimulated it in HT29 cells, whereas the ratio of pSTAT3 is presented by normalized to GAPDH.

**Figure 2 ijms-21-07315-f002:**
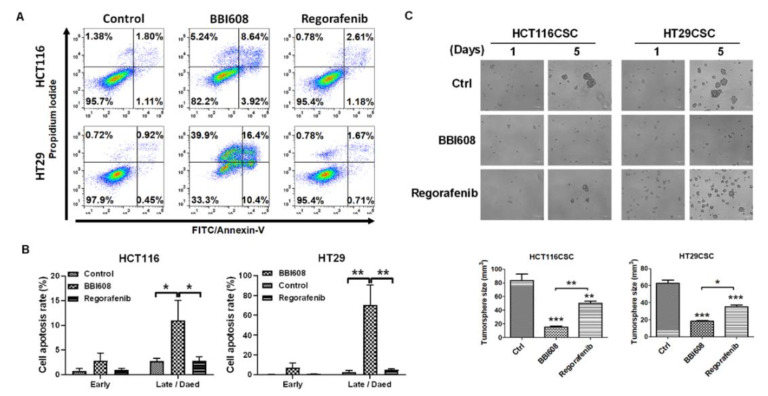
STAT3 inhibitor BBI608 prevented CRC-derived tumorspheres formation. (**A**) BBI608 induced stronger apoptotic effects than regorafenib in HCT116 and HT29 cells. (**B**) Quantification of apoptosis. (**C**) BBI608 significantly reduced the sizes of HCT116- and HT29-derived tumorspheres compared with controls or regorafenib-treated cells. Each IC50 concentration of BBI608 and regorafenib was used for detecting apoptosis and tumorsphere formation. Bar scale: 100 μm. * *p* < 0.01, ** *p* < 0.01, *** *p* < 0.001.

**Figure 3 ijms-21-07315-f003:**
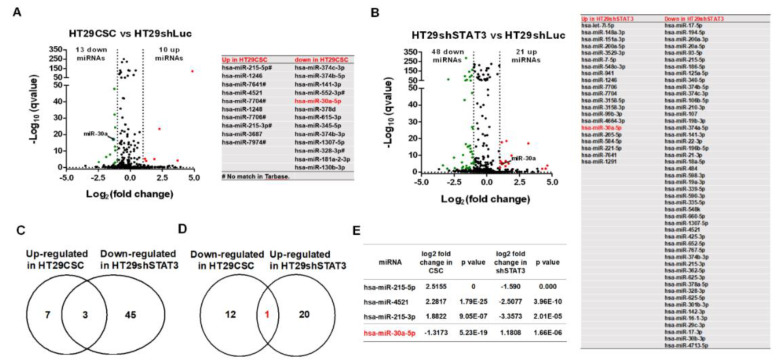
STAT3-mediated miRNA expression profile in HT29CSC revealed miR-30a-5p as a potential tumor suppression regulator. (**A**,**B**) A miRNA-seq was used to search and validate differential miRNAs involved in tumorsphere formation. A total of 10 of 21 miRNAs were upregulated and 13 of 48 miRNAs were downregulated in HT29CSC (**A**) and HT29shSTAT3 (**B**) cells. (**C**–**E**) Comparison of miRNAs profiles between HT29CSC and HT29shSTAT3 cells revealed that three miRNAs (miR-251-5p, miR-4251, miR-251-3p) were upregulated in HT29CSC cells (**C**,**E**) but downregulated in HT29shSTAT3 cells (**D**,**E**). By contrast, only miR-30a-5p was downregulated in HT29CSC cells but upregulated in HT29shSTAT3 cells (**E**).

**Figure 4 ijms-21-07315-f004:**
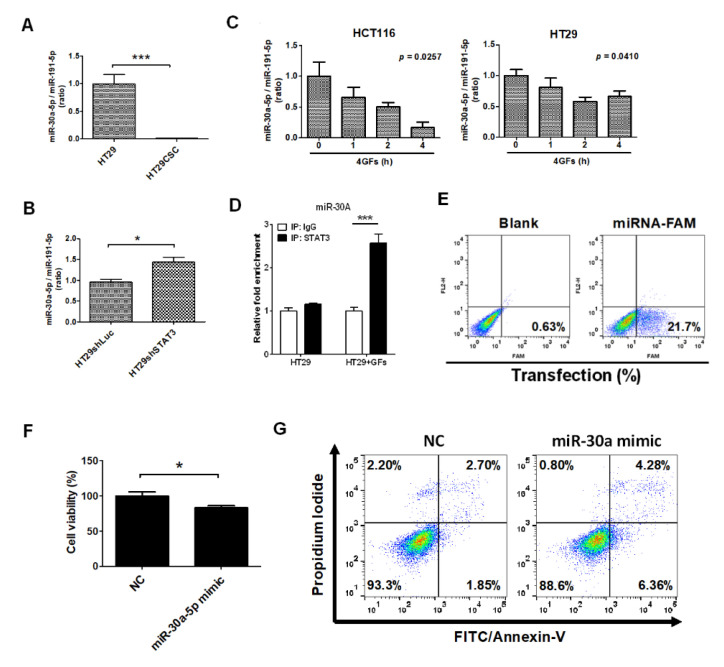
STAT3 downregulated miR-30a expression in CRC-derived stem-like cells. (**A**) miR-30a-5p levels were significantly decreased in HT29CSC (**B**) but increased in HT29shSTAT3 cells. (**C**) Treatment with 20 ng/mL of combined EGF, FGF, HGF, and IL-6 significantly reduced the miR-30a expression level in HCT116 and HT29 cells. (**D**) ChIP analysis indicated that STAT3 targeted the promoter of miR-30a. (**E**) Flow cytometry analysis of HT29 cells with miR-30a-5p transfection. miR-30a significantly reduced cell viability (**F**) and promoted cell apoptosis (**G**) at a transfection efficiency of < 21.7%. * *p* < 0.01, *** *p* < 0.001.

**Figure 5 ijms-21-07315-f005:**
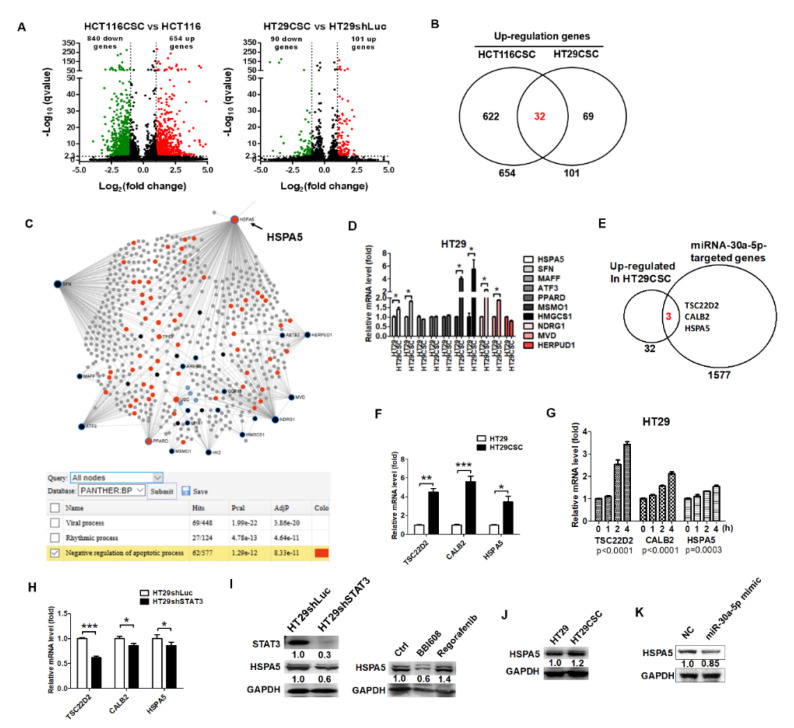
Gene expression profile in HCT116CSC and HT29CSC cells revealed that STAT3-mediated miR30a-5p downregulation increased HSPA5 expression to promote cancer stemness. (**A**) RNAseq data revealed 654 and 101 upregulated genes and 840 and 90 downregulated genes in HCT116CSC and HT29CSC cells, respectively. Among them, (**B**) there were 32 common genes upregulated in both HT116CSC and HT29CSC cells. (**C**) NetworkAnalyst analysis identified 10 genes (*HSPA5*, *SFN*, *MAFF*, *ATF3*, *PPARD*, *MSMO1*, *HMGCS1*, *NDRG1*, *MCD*, and *HERPUD1*) as key regulators, whereas *HSPA5* was associated with anti-apoptosis. (**D**) The expression levels of the 10 genes were validated by qPCR. HSPA5, SFN, MSMO1, HMGCS1, NDRG1, and MCD were significantly upregulated in HT29CSC cells compared with HT29 cells. (**E**) Following comparisons between RNAseq data and upregulated genes in HT29CSC cells, miR-30a was predicted to bind to three genes, namely *TSC22D2*, *CALB2*, and *HSPA5*. (**F**) The expression levels of the three genes were validated and found to be overexpressed in HT29CSC cells compared with HT29 cells (**G**) and growth factor-treated HT29 cells compared with non-treated cells; (**H**) however, the levels were decreased in HT29shSTAT3 cells compared with HT29shLuc cells. (**I**) The expression of HSPA5 was further detected using Western blotting. Significant decreases in expression levels were observed in HT29shSTAT3 compared with HT29shSTAT3 cells, and in BBI608-treated HT29 cells. (**J**) Moreover, HSPA5 increased in HT29-derived tumorspheres compared to parental HT29 cells, (**K**) but it decreased in miR-30a-5p mimic-transfected HT29 cells, thus demonstrating that STAT3-miR-30a increased HSPA5 expression. The protein levels normalized to GAPDH were quantified using ImageJ software. * *p* < 0.01, ** *p* < 0.01, *** *p* < 0.001.

**Figure 6 ijms-21-07315-f006:**
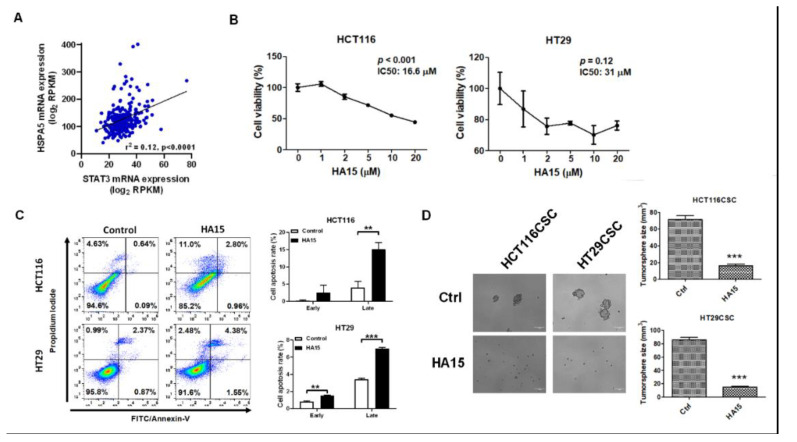
HSPA5 blockade inhibited cancer stemness in CRC cell lines. (**A**) Gene expression correlation of STAT3 and HSPA5 using RNAseq datasets from cBioPortal. Pearson’s correlation coefficient: r = 0.35. (**B**) The HSPA5 inhibitor HA15 significantly reduced cell viability and (**C**) induced cell apoptosis in HCT116 and HT29 cells. (**D**) Moreover, HA15 inhibited tumorsphere formation in HCT116 and HT29 cells. IC50 concentration of HA15 was used for detecting apoptosis and tumorsphere formation. *** *p* < 0.001.

**Table 1 ijms-21-07315-t001:** Primer sequence for qPCR.

Genes		Primer Sequence
*HSPA5*	Forward	TAGCGTATGGTGCTGCTGTC
	Reverse	TTTGTCAGGGGTCTTTCACC
*SFN*	Forward	AGAGCGAAACCTGCTCTCAG
	Reverse	CTCCTTGATGAGGTGGCTGT
*MAFF*	Forward	TCTGTGGATCCCCTATCCAG
	Reverse	CTTCTGCTTCTGCAGCTCCT
*ATF3*	Forward	GTGCCGAAACAAGAAGAAGG
	Reverse	TGGAGTCCTCCCATTCTGAG
*PPARD*	Forward	ACTGAGTTCGCCAAGAGCAT
	Reverse	GCGTTGAACTTGACAGCAAA
*MSMO1*	Forward	ATCCAGCTGCCTTTGATTTG
	Reverse	TTCCAAATGGAGCCTGAAAC
*HMGCS1*	Forward	CAAAAAGATCCATGCCCAGT
	Reverse	AAAGGCTTCCAGGCCACTAT
*NDRG1*	Forward	ACAACCCTGAGATGGTGGAG
	Reverse	TGTGGACCACTTCCACGTTA
*MVD*	Forward	AGGACAGCAACCAGTTCCAC
	Reverse	GTGTCGTCCAGGGTGAAGAT
*HERPUD1*	Forward	ACTTGCTTCCAAAGCAGGAA
	Reverse	CCCTTTGCCTTAAACCATCA
*TSC22D2*	Forward	GCAGCGTCCTGACTAGATCC
	Reverse	GAGCTGTCAGTTCCCGACTC
*CALB2*	Forward	GCTCCAGGAATACACCCAAA
	Reverse	CAGCTCATGCTCGTCAATGT
*GAPDH*	Forward	GAGTCAACGGATTTGGTCGT
	Reverse	TTGATTTTGGAGGGATCTCG
ChIP-*miR-30a-5p*	Forward	CACACCCACTTTTCTGTATCAACT
	Reverse	GAATTCCACTCCCATTCTCTTATG

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
