# Peer review of "STAT3 Mediated miR-30a-5p Inhibition Enhances Proliferation and Inhibits Apoptosis in Colorectal Cancer Cells"

_ijms, 2020, doi:10.3390/ijms21197315_

Round 1

Reviewer 1 Report

In this manuscript, the authors studied STAT3-mediated miRNAs and their roles in tumorsphere formation and tumor apoptosis in colorectal cancer. The results showed that four STAT3 mediated miRNAs were identified by RNAseq: three were upregulated and one (miR-30a-5p) was down-regulated during tumorsphere formation. This study also showed that HSP5A is the downstream target of miR-30a-5p. Knockdown of STAT3 by shRNA increased the expression of miR-30a-5p and reduced the expression of HSP-5A. Inhibition of STAT3 and HSPA5 with STAT3 inhibitor (BBI608) and HSPA5  inhibitor (HA15), respectively, led to increased tumor apoptosis and decreased tumorsphere formation. The topic is important and can be published after the paper is revised. 

  1. Abstract: can be improved for clarity. The significance and novelty should be pointed out as well
  2. Figures: Figures are blurry and too small. The axes and some labels can’t be recognized.
  3. Concentrations: molar concentration, which is more commonly used, should be used or included.
  4. Material and methods: the sources of the reagents,such as BBI608, HA15 and antibodies, should be included.  More information should be included in each experiment.
  5. Experimental conditions and details, such as concentration, should be described in each experiment. Some patient information should be described for the Kaplan-Meier plotter. The p-values for Kaplan-Meier plotter is larger than 0.05, which is not significant and should be discussed.
  6. Apoptosis data: Bar graph with error bar also should be included. Experimental details should be described.
  7. Figure 4F and 4G: the results are not convincing enough to support the conclusion. This could be due to low transfection rate. The results should be discussed. Any effect of miR-30a-5p overexpression on tumorsphere formation?
  8. Figure 4: what is the effect of STAT3 knockdown on tumorsphere formation and apoptosis?
  9. The expression of each band in Western blot should be quantified by densitometry.

Author Response

1. Abstract: can be improved for clarity. The significance and novelty should be pointed out as well

Response: We appreciate the reviewer’s comment. We point out our finding and revise as “In conclusion, a STAT3-miR-30a-5p-HSPA5 axis was observed against ragorafenib-mediated apoptosis in colorectal cancer tumorspheres. The expression of miR-30a-5p was repressed by STAT3; in addition, HSPA5 was identified as the target gene of miR-30a-5p and contributed to both tumorsphere formation and anti-apoptosis.” in Abstract.

2. Figures: Figures are blurry and too small. The axes and some labels can’t be recognized.

Response: We appreciate the reviewer’s comment. We re-draw the Figures containing apoptosis data and RNAseq data for improving the resolution, including Figure 2A, 3A, 3B, 4E, 4G, 5A, and 6C.

3. Concentrations: molar concentration, which is more commonly used, should be used or included.

Response: We appreciate the reviewer’s comment. We change the unit by molar concentration in 2A and 6B and in the revised manuscript.

4. Material and methods: the sources of the reagents, such as BBI608, HA15 and antibodies, should be included.  More information should be included in each experiment.

Response: We appreciate the comment. We added and described as “The reagents, including BBI608, regorafenib, and HA15 (MedChemExpress, Monmouth Junction, NJ, USA) were used and added in HCT116 and HT29 cells for 48 h.” in section 4.7. Moreover, we added the details about mRNA extraction, cDNA preparation, gene knockdown, western blots, and ChIP in Materials and Methods.

5. Experimental conditions and details, such as concentration, should be described in each experiment. Some patient information should be described for the Kaplan-Meier plotter. The p-values for Kaplan-Meier plotter is larger than 0.05, which is not significant and should be discussed.

Response: We appreciate the constructive comment. The details are added in the figure legend as “Each IC50 concentration of BBI608 and regorafenib was used for detecting apoptosis and tumorsphere formation.” in Figure 2, and “IC50 concentration of HA15 was used for detecting apoptosis and tumorsphere formation.” in Figure 6. In addition, the Kaplan-Meier plotter dataset (https://kmplot.com) was used that we added patient number in the revised manuscript.  Although the p value larger than 0.05 but it shows the trend associating with survival probability in patients with rectum adenocarcinoma. We add the description in Discussion.

6. Apoptosis data: Bar graph with error bar also should be included. Experimental details should be described.

Response: We appreciate the constructive comment. We added the bar graph showing apoptosis detection in 2B and 6C. The experimental details are added and described in section 4.10 flow cytometry as “For apoptosis detection, a commercial kit containing Annexin V-FITC and Propidium Iodide was used (Strong Biotech Corporation, Taiwan). The tumor cells were treated with IC50 of inhibitor for 24 h, including BBI608, regorafenib, and HA15. HT29 cells transfected with miR-30a-5p mimic and negative control sequence were collected after 24 h. The cells were analyzed using a FACSCalibur Flow Cytometer (BD Bioscience, CA, USA) after Annexin V-FITC and Propidium Iodide staining.”

7. Figure 4F and 4G: the results are not convincing enough to support the conclusion. This could be due to low transfection rate. The results should be discussed. Any effect of miR-30a-5p overexpression on tumorsphere formation?

Response: We appreciate the comment. miR-30a-5p mimic was transfected into HT29 cells, it decreased cell viability in Figure 4F, and increased apoptosis in Figure 4G. The result is as expected since miR-30a-5p performed as a tumor suppressor. This study found that STAT3 was able to suppress miR-30a-5p, resulting in increase of HSPA5. 

8. Figure 4: what is the effect of STAT3 knockdown on tumorsphere formation and apoptosis?

Response: We appreciate the constructive comment. We previously found that knockdown of STAT3 decreased tumorsphere formation and reduced cell viability in HT29 cells (Cheng et al. STAT3 exacerbates survival of cancer stem-like tumorspheres in EGFR-positive colorectal cancers: RNAseq analysis and therapeutic screening. Journal of Biomedical Science (2018) 25:60). Since the HT29shSTAT3 is a stable cell line, we therefore did not detect apoptosis. According to the data performing STAT3 inhibitor BBI608, inhibition of STAT3 led to significant apoptosis in HCT116 and HT29 cells (Figure 2B).

9. The expression of each band in Western blot should be quantified by densitometry.

Response: We appreciate the comment. The Western blots were quantified using ImageJ software.

Reviewer 2 Report

The manuscript by Cheng, Yang and Chen et al provided interesting insight and shows strong correlation between STAT3, miRNA and UPR in colorectal cancer. The authors used two inhibitors to establish differential effect of the inhibitors on 2D vs 3D tumorsphere model. Regorafenib is an inhibitor for VEGFRs, Ret, Kit, PDGFR and Raf kinases, which can result in the inhibition of tumor angiogenesis and tumor cell proliferation. On the other hand, napabucasin or BBI608 is a cancer stemness inhibitor which appears to target and inhibit multiple pathways involved in cancer cell stemness. This may ultimately inhibit cancer stemness cell growth as well as heterogeneous cancer cell growth. While the manuscript is STAT3 centric, the authors failed to establish the connection between these inhibitors and the reason to use them in first place and that requires justification. This reviewer has some suggestions for improving the manuscript in its current format before recommending it for acceptance.

  • Figure 1: The authors should justify why CD133 and LGR5 expression was assessed and their relevance with regards to Regorafenib and STAT3 inhibitor use.
  • If anything, the authors can evaluate changes in CD133 and LGR5 expression between 2D and 3D tumorspheres following treatment with BBI1608 and Regorafenib.
  • Since the paper is STAT3 centric. It would add to the flow if the authors could show endogenous levels of phosphorylated STAT3 and total STAT3 in both the colorectal cancer cell lines before and after inhibitors treatment at their respective IC50 doses.
  • In addition, the authors should determine levels of activated and total STAT3 in 2D cell line vs 3D tumorspheres. This could explain differential trends in miRNA between CSC and shSTAT3 knockdown.
  • As a suggestion can the authors show the effect of STAT3 inhibition on tumorsphere growth using another ‘more-specific’ STAT3 inhibitor as either of the inhibitors used are not specific for STAT3.
  • Figure 2: The labelling and gray color scheme is not legible for 2B. Can the authors please make it black and increase font size across all the bar graphs in 2A, B and C.
  • Figure 2C: Authors claim that tumorspheres formed by the 2 cancer cell lines were resistant to regorafenib. However, there is a clear and consistent reduction in tumorspheres viability supported by p value. As a result, this reviewer would suggest that the authors re-phrase the 2.2 section heading to state that the tumorspheres were more sensitive to BBI608.
  • Regarding Tumorsphere formation the authors should briefly describe how they are formed and how/what viability assay was performed. In the current report they have cited Ref#37 which is on lung cancer cell lines while the manuscript under review is on colorectal. The readers would benefit if that information was available within the same manuscript.
  • Figure 3F: While this reviewer appreciates Kaplan Meier curves shown here but none of the curves achieved significance and therefore can be referred to as trend/correlatives/associations if they are not significant. Therefore, the authors should reword this in Results and Discussion. In addition, this should be moved to supplementary. Alternatively, the authors can investigate is there is correlation in TCGA data between the different miRNA and STAT3 and or HSPA5 to replace survival curves.
  • Can authors justify why HSPA5 was chosen as a downstream target. Was HSPA5 the only anti-apoptotic molecule that was differentially regulated?
  • Figure 5J: Please quantify or repeat the blot as it is difficult to establish an increase in HSPA5.
  • The authors should change “Tumor size” to “Tumorsphere size” across all the graphs.
  • Can authors test if BBI608 and Regorafenib treatment on Tumorspheres reduced HSPA5 and to a differential extent to support Figure 6 and differences observed in Figure 2C.
  • The title could be re-worded. A possible suggestion is “STAT3 mediated miR-30a-5p inhibition enhances proliferation and inhibits apoptosis in colorectal cancer cells”.

Author Response

  • Figure 1: The authors should justify why CD133 and LGR5 expression was assessed and their relevance with regards to Regorafenib and STAT3 inhibitor use.

Response: We appreciate the comment. We have investigated the CD133 level in BBI608 or regorafenib treatment (only detect CD133 because HCT116 and HT29 are CD133-positive) but there is no significant change in HT29. In this study, we demonstrate the CD133 and LGR5 expression because these two proteins are stemness markers. Since there is no correlation between CD133 and inhibitors, we delete the data and put on as supplemental data (Figure S1).

  • If anything, the authors can evaluate changes in CD133 and LGR5 expression between 2D and 3D tumorspheres following treatment with BBI1608 and Regorafenib.

Response: We appreciate the comment. We previously demonstrated that CD133 and LGR5 increase in HT29-derived tumorsphere (Cheng, C.C., Hsu, P.J., Sie, Z.L., Chen, F.H. Discovery of Driver Genes in Colorectal HT29-derived Cancer Stem-Like Tumorspheres. J. Vis. Exp. e61077, doi:10.3791/61077 (2020)). But, there is no significant increase of CD133 in HCT116-derived tumorspheres because HCT116 cells are positive by 97.5%. In addition, there is no correlation between CD133 and the inhibitors (BBI1608 and Regorafenib), therefore, we delete the data and put on as supplemental data.

  • Since the paper is STAT3 centric. It would add to the flow if the authors could show endogenous levels of phosphorylated STAT3 and total STAT3 in both the colorectal cancer cell lines before and after inhibitors treatment at their respective IC50 doses.

Response: We appreciate the constructive comment. The endogenous pSTAT3 and total STAT3 in HT29 after inhibitor treatment are added in Figure 1B. It shows that BBI608 dramatically inhibited endogenous pSTAT3 but regorafenib increased pSTAT3 in HT29 cells.

  • In addition, the authors should determine levels of activated and total STAT3 in 2D cell line vs 3D tumorspheres. This could explain differential trends in miRNA between CSC and shSTAT3 knockdown.

Response: We appreciate the constructive comment. The four growth factors including EGF, bFGF, IL6, and HGF were used for culture tumorspheres. EGF and IL6 are demonstrated to phosphorylate STAT3 in cancer cells (Reference 8, 12, 18). Since pSTAT3 always appears in 15 mins after adding growth factors, but the tumorspheres were cultured for 7 days, we therefore prefer not show pSTAT3 between tumorspheres and the parental cells.

  • As a suggestion can the authors show the effect of STAT3 inhibition on tumorsphere growth using another ‘more-specific’ STAT3 inhibitor as either of the inhibitors used are not specific for STAT3.

Response: We appreciate the constructive comment. Our previous study demonstrates that another STAT3 inhibitor homoarringtonin blocks STAT3 expression and also reduces the T29-derived tumorsphere formation. (Cheng et al. STAT3 exacerbates survival of cancer stem-like tumorspheres in EGFR-positive colorectal cancers: RNAseq analysis and therapeutic screening. Journal of Biomedical Science (2018) 25:60). Moreover, in the previous study we reveal knockdown of STAT3 decreased tumorsphere formation and reduced cell viability in HT29 cells

  • Figure 2: The labelling and gray color scheme is not legible for 2B. Can the authors please make it black and increase font size across all the bar graphs in 2A, B and C.

Response: We appreciate the reviewer’s comment. We re-draw the Figures containing apoptosis data and RNAseq data, including Figure 2B, 3A, 3B, 4E, 4G, 5A, and 6C. Moreover, we increase font size for the bar graphs in this study.

  • Figure 2C: Authors claim that tumorspheres formed by the 2 cancer cell lines were resistant to regorafenib. However, there is a clear and consistent reduction in tumorspheres viability supported by p value. As a result, this reviewer would suggest that the authors re-phrase the 2.2 section heading to state that the tumorspheres were more sensitive to BBI608.

Response: We appreciate the reviewer’s comment. We change the description as “HCT116- and HT29-derived tumorspheres were more sensitized to BBI608 than regorafenib” according to the comment. Thanks a lot.

  • Regarding Tumorsphere formation the authors should briefly describe how they are formed and how/what viability assay was performed. In the current report they have cited Ref#37 which is on lung cancer cell lines while the manuscript under review is on colorectal. The readers would benefit if that information was available within the same manuscript.

Response: We appreciate the reviewer’s comment. We add as “In this study, EGF, bFGF, IL6, and HGF (GeneScript, Piscataway, New Jersey, USA) were used to culture tumorspheres for 7-days in low-attachment plate without fetal bovine serum. The diameter of the tumorsphere was recorded through an inverted microscope.” in 4.1. and The reagents, including BBI608, regorafenib, and HA15 (MedChemExpress, Monmouth Junction, NJ, USA) were used and added in HCT116 and HT29 cells for 48 h for cell viability detection.” in 4.7.

  • Figure 3F: While this reviewer appreciates Kaplan Meier curves shown here but none of the curves achieved significance and therefore can be referred to as trend/correlatives/associations if they are not significant. Therefore, the authors should reword this in Results and Discussion. In addition, this should be moved to supplementary. Alternatively, the authors can investigate is there is correlation in TCGA data between the different miRNA and STAT3 and or HSPA5 to replace survival curves.

Response: We appreciate the reviewer’s comment. The Kaplan Meier curves are moved to Figure S2 and Figure S3 as supplemental data because no statistical significant. But it still reveals a trend for the gene and survival rate in patients with rectum adenocarcinoma. In addition, we add the correlation between STAT3 and HSPA5 in Figure 6A. Thanks a lot.

  • Can authors justify why HSPA5 was chosen as a downstream target. Was HSPA5 the only anti-apoptotic molecule that was differentially regulated?

Response: We appreciate the reviewer’s comment. We add the description in Result 2.5 as “Since HSPA5 was associated with anti-apoptosis, as predicted by NetworkAnalyst.”

  • Figure 5J: Please quantify or repeat the blot as it is difficult to establish an increase in HSPA5.

Response: We appreciate the reviewer’s comment. We quantify the Western blot data in this study using ImageJ software. Thanks a lot.

  • The authors should change “Tumor size” to “Tumorsphere size” across all the graphs.

Response: We appreciate the reviewer’s comment. We change as “tumorsphere size” in the revised manuscript (Figure 2C and 6D).

  • Can authors test if BBI608 and Regorafenib treatment on Tumorspheres reduced HSPA5 and to a differential extent to support Figure 6 and differences observed in Figure 2C.

Response: We appreciate the reviewer’s comment. We add the data in the revised manuscript that only BBI608 reduced HSPA5 expression shown in in Figure 5I.

  • The title could be re-worded. A possible suggestion is “STAT3 mediated miR-30a-5p inhibition enhances proliferation and inhibits apoptosis in colorectal cancer cells”.

Response: Many thankful for the reviewer’s comment. The title is changed as the comment.

Reviewer 3 Report

Dear Authors,

The study aims to elucidate the role of STAT3 in the development of drug resistance by CRC cells to regorafenib. Also, STAT3-mediated miRNAs and their role in drug resistance was evaluated.

I found the study not to be very well presented. Grammar and language also require improvement.

It is difficult to follow the text – the aim/purpose of all the experiments/assays should be clarified. Moreover, all the data should be described in more details.

Below, please find a few examples of many other issues:

  1. Introduction – can you comment on how many CRC patients received regorafenib treatment and how many of them developed resistance to the drug?
  2. Section 2.1 - What is the purpose of presenting this data as an independent section. The text should be shortened to two sentences and figures should be placed in the Supplementary file.
  3. Fig 2A - IC50 study; the Authors used 0.1, 1 and 10 ug/mL (10x factor) concentrations of each of the compounds. Did you test any other concentrations of inhibitors (0.2 or 0.5 ug/mL)? Can you present IC50 data in a standard format - as a linear graph (not column).
  4. Please provide better quality of all cytograms (Fig. 2B, 4G etc.)
  5. “HCT116 and HT29 cells gifted by the Institute of Nuclear Energy Research, Taiwan were free of 272 mycoplasma” – the source of the cell lines is not clear – what is the primary source of cells?

The entire section o M&M is not informative. There is significant lack of important information such as: cell growth conditions (lack or presence of FBS and in which variant, etc?), the used concentrations of antibodies, description of tumorsphere formation assay, apoptosis assay, etc.  

The primers used in the experiment should be listed in a table.

Concluding, I cannot recommend the manuscript for publication in the Journal in the present form.

Author Response

The study aims to elucidate the role of STAT3 in the development of drug resistance by CRC cells to regorafenib. Also, STAT3-mediated miRNAs and their role in drug resistance was evaluated.

I found the study not to be very well presented. Grammar and language also require improvement.

Response: We appreciate the reviewer’s comment. The manuscript was edited by Wallace Academic Editing for improving English editing.

It is difficult to follow the text – the aim/purpose of all the experiments/assays should be clarified. Moreover, all the data should be described in more details.

Response: We appreciate the reviewer’s constructive comment. We rewrite our finding and revise as “In conclusion, a STAT3-miR-30a-5p-HSPA5 axis was observed in colorectal cancer tumorspheres against regorafenib-mediated apoptosis. The expression of miR-30a-5p was repressed by STAT3; in addition, HSPA5 was identified as the target gene of miR-30a-5p and contributed to both tumorsphere formation and anti-apoptosis” in Abstract. Moreover, more detail about methodology is shown in Methods and Materials. Many thanks for your comments.

Below, please find a few examples of many other issues:

1. Introduction – can you comment on how many CRC patients received regorafenib treatment and how many of them developed resistance to the drug?

Response: We appreciate the comment. We cite the disease control rate from a previous study in Introduction, and describe as “Regorafenib has been approved for use in patients with metastatic colorectal cancer, resulting in 26% disease control rate (complete response, partial response, or stable disease lasting at least 6 weeks) [2] , but the antitumor activity of regorafenib is restricted to intrinsic and acquired resistance in CRC.”

2. Section 2.1 - What is the purpose of presenting this data as an independent section. The text should be shortened to two sentences and figures should be placed in the Supplementary file.

Response: We appreciate the comment. We have investigated the CD133 level in BBI608 or regorafenib treatment but there is no significant change in HT29 (data not shown). In this study, we demonstrate the CD133 or LGR5 expression because these two proteins are stemness markers. Since there is no correlation between CD133 and inhibitors, we delete the data and move as supplemental data (Figure S1).

3. Fig 2A - IC50 study; the Authors used 0.1, 1 and 10 ug/mL (10x factor) concentrations of each of the compounds. Did you test any other concentrations of inhibitors (0.2 or 0.5 ug/mL)? Can you present IC50 data in a standard format - as a linear graph (not column).

Response: We appreciate the reviewer’s constructive comment. We have tested 0, 1.25, 2.5, 5, and 10 μg/mL of regorafenib to calculate IC50. The result was similar. In addition, we change the IC50 data using linear graph according to your comment in Figure 2A and 6B.

4. Please provide better quality of all cytograms (Fig. 2B, 4G etc.)

Response: We appreciate the reviewer’s constructive comment. We re-draw the Figures containing apoptosis data and RNAseq data for improving the blurry labels, including Figure 2B, 3A, 3B, 4E, 4G, 5A, and 6C.

5. “HCT116 and HT29 cells gifted by the Institute of Nuclear Energy Research, Taiwan were free of 272 mycoplasma” – the source of the cell lines is not clear – what is the primary source of cells?

Response: We appreciate the reviewer’s constructive comment. We revised as “HCT116 and HT29 cells gifted by the Institute of Nuclear Energy Research, Taiwan were free of mycoplasma, that were originally purchased from ATCC.”

The entire section o M&M is not informative. There is significant lack of important information such as: cell growth conditions (lack or presence of FBS and in which variant, etc?), the used concentrations of antibodies, description of tumorsphere formation assay, apoptosis assay, etc.  

Response: We appreciate the reviewer’s constructive comment. More details are added in Methods and Materials in the revised manuscript.

The primers used in the experiment should be listed in a table.

Response: We appreciate the reviewer’s constructive comment. The primers are listed in Table 1.

Round 2

Reviewer 1 Report

The authors have addressed my concerns in the revised manuscript.

Author Response

The authors have addressed my concerns in the revised manuscript.

Response: Many thankful for your review and comments.

Reviewer 2 Report

I thank the reviewers for making the effort of incorporating the suggested edits.

I only have minor concerns:

Figure 1: The authors should state in corresponding legend what the densitometry is for? Is it pSTAT3 or total STAT3?

Figure 1: The authors should verify the densitometry for the blot showing Regorafenib treatment  on HT29 cells. What is pSTAT3 being normalized against GAPDH or total STAT3 considering the loading is not the same across the 24h HT29 blot?

Author Response

Figure 1: The authors should state in corresponding legend what the densitometry is for? Is it pSTAT3 or total STAT3?

Response: We appreciate your constructive comment. We add the statement as “whereas ratio of pSTAT3 is presented by normalized to GAPDH.” Moreover, we add “The protein levels normalized to GAPDH were quantified using ImageJ software” in Figure 5.

Figure 1: The authors should verify the densitometry for the blot showing Regorafenib treatment on HT29 cells. What is pSTAT3 being normalized against GAPDH or total STAT3 considering the loading is not the same across the 24h HT29 blot?

Response: We appreciate your constructive comment. We add the statement indicating that ratio of pSTAT3 is presented by normalized to GAPDH in the legend of Figure 1. According to the result, BBI608 inhibited intrinsic pSTAT3 (2 h treatment) in HT29 cells, resulting in decrease of STAT3 levels (24 h treatment). The decrease of protein levels was because the cells were being apoptosis (Figure 2A).

Reviewer 3 Report

Dear Authors,

Thank you for provided changes. I will recommend manuscript for publication.

Author Response

Thank you for provided changes. I will recommend manuscript for publication.

Response: Many thankful for your review and comments.